# Large-Eddy Simulation of Flow Separation Control in Low-Speed Diffuser Cascade with Splitter Blades

Zhong Liang [1], Jun Wang [1,*], Boyan Jiang [1], Hao Zhou [1], Weigang Yang [2] and Jieda Ling [2]

1   School of Energy and Power Engineering, Huazhong University of Science and Technology, Wuhan 430074, China; zliang_hust@163.com (Z.L.); jiangby_hust@163.com (B.J.); zhouhao199609@163.com (H.Z.)
2   Ningbo Fotile Kitchen Ware Co., Ltd., Ningbo 315000, China; yangwg@fotile.com (W.Y.); lingjd@fotile.com (J.L.)
*   Correspondence: wangjhust@hust.edu.cn; Tel.: +86-027-87542517-85

**Abstract:** The passive flow control technology of using splitter blades in low-speed diffuser cascade was investigated in this study. Based on the Reynolds average Navier-Stokes calculations, the arrangement parameters of the splitter blades were studied in detail to determine the optimal parameters. The large-eddy simulation was performed on the base case and the optimized splitter blade case to obtain the transient vortex structures and unsteady flow characteristics of the cascade. The results show that the aerodynamic performance of the cascade was susceptible to the position of the splitter blades. The optimal position of the splitter blades was located in the middle of the main blades near the leading edge. When the cascade was arranged with optimized splitter blades, the static pressure coefficient was improved and the stall occurrence was delayed. The scale and intensity of the separation vortices generated on the suction surface of the main blade decreased. In addition, the separation vortices of the main blade and the splitter blade interacted and rapidly decomposed into small-scale vortices downstream of the cascade, reducing the flow loss. The stability of the cascade was enhanced.

**Keywords:** passive flow control; splitter blades; low-speed diffuser cascade; large-eddy simulation

## 1. Introduction

Flow separation on axial flow fan blades usually occurs at low-flow-rate conditions [1]. Under a large adverse pressure gradient, the flow separation occurs on the suction side of the blade with the increase in the attack angle. The separation area increases, eventually blocking the flow passage, and the rotating stall occurs [2]. When the stall occurs, the aerodynamic performance of the fan will deteriorate, intense noise will be induced, and reliability will be reduced [3]. Therefore, it is a research hotspot in the field of turbomachinery to explore the flow separation mechanism of axial flow fans at high angles of attack and seek effective control technology [4].

It is of great realistic significance to manipulate the flow field to achieve the expected changes [5]. According to energy consumption, the current flow control techniques are mainly divided into two categories: active flow control technology, which involves energy and requires a control system, and passive flow control technology, which requires no energy consumption [6]. With the development of computational fluid dynamics methods in the past three decades, the feasibility of active flow control technology has been dramatically improved. Ortiz-Tarin [7,8] used a hybrid simulation to investigate the wake of a slender body and found that there is no universal framework for the wake. Xu [9,10] found that the mixing of boundary layers is promoted by the periodic flow field disturbance introduced by the synthetic jet. The enhancement of the momentum transport near the boundary layers and the increase in the inverse pressure gradient resistance of the primary vortex retard the flow separation process. Zhu [11] adopted jet actuators with

an upward-parabola blowing control strategy to conspicuously improve the aerodynamic performance of the vertical axis wind turbines with significantly less material and energy consumption. Kamari [12] compared boundary layer blowing and suction. The results show that both flow control methods can inhibit flow separation, but suction is more effective than blowing.

Although active flow control technology has extensive adaptability, passive flow control technology is simpler and easier to implement. In addition, passive flow control technology started earlier and has been more widely used in engineering [6]. Brüder-lin [13] investigated the effectiveness of passive vortex generators with different dimensions under a series of flow conditions and deflection angles. Yang [14] adopted riblets to improve the aerodynamic performance of the airfoil. The "antifriction-bearing" structure formed by riblets on the airfoil surface effectively suppressed the trailing separation vortex. Wang [15,16] studied the leading-edge slat with different geometric parameters on the aerodynamic performance of the S809 airfoil. It was shown that the leading-edge slat could delay the flow separation and increase the lift coefficient. Mostafa [17] increased the power output of the horizontal axis wind turbine blade by adding a micro-cylinder.

As a passive flow control technology, the splitter blade adds a reduced chord airfoil in the passage defined by the main airfoils. This technology is widely used in centrifugal turbomachinery to improve pressure ratio and efficiency. Malik [18,19] improved the pressure ratio and efficiency by adding multi-splitter blades in the centrifugal compressor impeller. Zhang [20] analyzed the effects of the splitter blade deflection on the performance and pressure fluctuation in the pump. Jia [21] designed a Francis turbine with splitter blades, which improves efficiency and reduces high-amplitude pressure fluctuation.

The application of splitter blades in axial flow turbomachinery started late. In 1974, Wennerstrom [22] first proposed installing splitter blades on the rotor of a supersonic axial compressor stage. Tzuoo [23] proposed an effective method for the design of a high-pressure ratio splitter impeller. The methods of deviation distribution, loss, solidity, specifying work distribution, and splitter blade arrangement were discussed. Li [24] simulated the unsteady flow fields of a single-stage axial compressor rotor with splitter blades under design conditions. The results showed that the splitter blades change the rotor channel's pressure distribution and the rotor's load distribution. The unstable static pressure distribution and fluctuation in the rotor were also reduced. Clark [25] adopted splitter blades to reduce the secondary flow intensity of low-aspect ratio vanes and increase stage efficiency. Liu [26] presented a model to predict the reference minimum-loss incidence and deviation angles of blade placement with splitter blades. This model provides a solution for the design of an ultra-high-loaded axial compressor. Pham [27] installed splitter blades to the stator of a single-stage axial compressor, effectively improving the operation stability. The splitter blades effectively restrained the flow separation in the stator at near-stall conditions. The above research on splitter blades in axial flow turbomachinery mainly focuses on transonic and high-load compressor rotors. There needs to be more research on the application of splitter blades in low-speed axial flow fans. As a universal turbomachinery, axial flow fans are widely used in various scenarios [28]. It is of great practical significance to control the flow separation and improve the efficiency of axial flow fans.

This study arranged the splitter blades in a low-speed diffuser cascade. The effects of the splitter blade on the aerodynamic performance and internal unsteady flow characteristics of the low-speed diffuser cascade were analyzed by numerical simulation. The flow control mechanism of splitter blades in the low-speed diffuser cascade was investigated. The study provides some reference for the application of splitter blades in low-speed axial flow fans.

## 2. Geometric Model and Splitter Blades

### 2.1. Low-Speed Diffuser Cascade

The low-speed diffuser cascade designed in reference [29] was selected as the research object in this article. The cascade adopted an airfoil of NACA65-010 (National Advisory Committee for Aeronautics, Columbia, WA, USA). The geometry and main parameters of the cascade are shown in Figure 1 and Table 1. The airfoil chord length ($c$) was 160 mm, and the blade span was 127 mm. The pitch spacing ($s$) of the main blade was 140 mm. The stagger angle of the cascade blade was 39°. The design attack angle (DAA) was 16°, and the stall attack angle (SAA) was 26°. The flow velocity ($U_{in}$) at the cascade inlet was 22 m/s, and the corresponding Mach number was 0.065. There were 12 static pressure measuring holes on the suction surface of 50% blade span to measure the blade surface pressure coefficient ($C_{ps}$). Moreover, the $C_{ps}$ was used to validate the numerical methods.

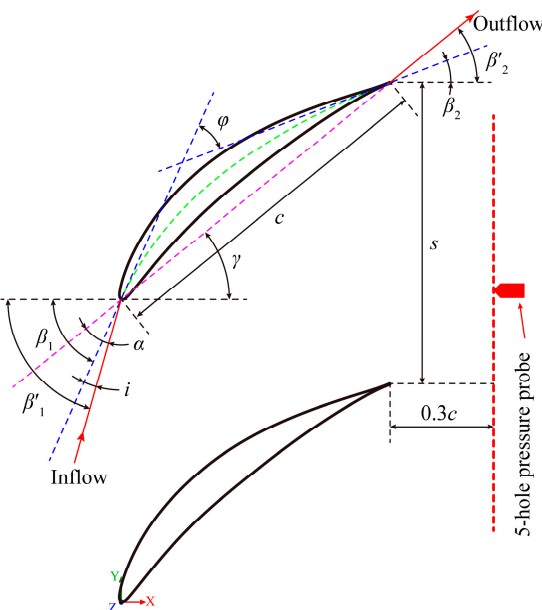

**Figure 1.** Sketch of the diffuser cascade parameters.

**Table 1.** Main geometrical parameters of the diffuser cascade.

| Parameter | Value |
| --- | --- |
| Chord $c$/mm | 160 |
| Camber angle $\varphi$/(°) | 45 |
| Stagger angle $\gamma$/(°) | 39 |
| Pitch spacing $s$/mm | 140 |
| Solidity $c/s$ | 1.143 |
| Blade span $h$/mm | 127 |
| Aspect ratio $h/c$ | 0.794 |
| Incidence angle $i$/(°) | 0~12 |
| Attack angle $\alpha$/(°) | 16~28 |
| Design inflow angle $\beta_1$/(°) | 55 |
| Design outflow angle $\beta_2$/(°) | 10 |
| Ma/(-) | 0.065 |

### 2.2. Splitter Blade Designs

There is no invariable design method for splitter blades at present. This study proposes a design method for splitter blades. The leading-edge point of the main blade was taken as the origin, and the *IJ* coordinate system was established in the direction of pitch and chord length. The airfoil of the splitter blade was obtained by reducing the main blade to 25%.

The reduced airfoil was moved first along the *I* direction and then along the *J* direction to obtain the splitter blade between the main blades.

As shown in Figure 2, 12 locations were selected to investigate the influence of splitter blade arrangement positions on the aerodynamic performance and internal unsteady flow state of the low-speed diffuser cascade. In the *I* direction, the distance between the splitter blade and the main blade was in the range of $I/c = 0 \sim 0.75$. It had an interval of $0.25c$ and is represented by a number ("1", "2", "3", or "4"), respectively. In the *J* direction, the splitter blade was located at $0.4s$, $0.5s$, and $0.6s$. Moreover, it is represented by a letter ("A", "B", or "C"), respectively. Each location of the splitter blade can be represented by a combination of a letter and a number. For example, A1 corresponds to the case ($I/c = 0$, $J/s = 0.4$). The corresponding coordinates of the splitter blades at 12 locations are listed in Table 2. The following chapters refer to the cascade without splitter blades as the "base case."

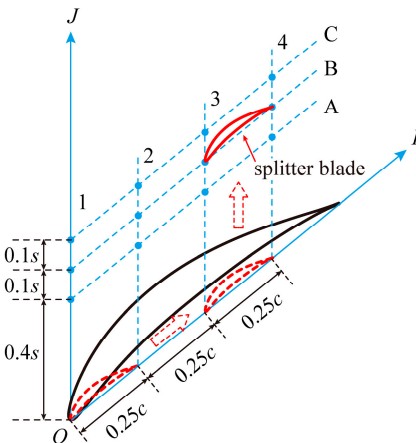

**Figure 2.** Schematic of the coordinate and position of the splitter blades.

**Table 2.** List the coordinates of the splitter blades in the *IJ* coordinate system.

| (*I/c*, *J/s*) | A | B | C |
|---|---|---|---|
| 1 | (0, 0.4) | (0, 0.5) | (0, 0.6) |
| 2 | (0.25, 0.4) | (0.25, 0.5) | (0.25, 0.6) |
| 3 | (0.5, 0.4) | (0.5, 0.5) | (0.5, 0.6) |
| 4 | (0.75, 0.4) | (0.75, 0.5) | (0.75, 0.6) |

## 3. Computational Methods and Validations

### 3.1. Numerical Models and Boundary Conditions

In this study, Reynolds averaged Navier-Stokes (RANS) and large-eddy simulation (LES) methods were implemented to study the effects of splitter blades on the aerodynamic performance and internal unsteady flow state of the low-speed diffuser cascade. A detailed study of the splitter blade position parameters was carried out based on the RANS calculation. The optimal position parameters for aerodynamic performance improvement and stall control of the cascade were obtained. The LES simulation can capture the delicate near-wall flow structures as well as the wake evolution [30,31]. The LES simulation of the base case and the optimized splitter blade case is supplemented to provide the transient flow field.

Since the Mach number of the inlet is 0.065, the flow in the cascade can be regarded as incompressible. The incompressible RANS simulation was performed in ANSYS FLUENT. The governing equations were as follows:

$$\frac{\partial \rho}{\partial t} + \frac{\partial \rho u_i}{\partial x_i} = 0 \tag{1}$$

$$\frac{\partial(\rho u_i)}{\partial t} + \frac{\partial}{\partial x_j}(\rho u_i u_j) = -\frac{\partial P}{\partial x_i} + \frac{\partial}{\partial x_j}(\mu \frac{\partial u_i}{\partial x_j} - \rho \overline{u_i' u_j'}) + S_i \tag{2}$$

where $u_i$ represents the velocity components averaged in the $x_i$ direction; and $\rho$, $\mu$, $t$, and $P$ are the air density, the fluid's dynamic viscosity, time, and average pressure, respectively.

The cascade is similar in the circumferential and spanwise directions, so the single-channel cascade model with a $0.15c$ span was adopted for calculation, as shown in Figure 3. This method effectively reduces the calculation cost. At the inlet of the cascade, the velocity inlet boundary condition was set. The airflow velocity at the inlet was 22 m/s. The outlet boundary was a pressure outlet with 0 Pa. The blade's end wall and circumferential wall were set as periodic surfaces, respectively. The $k$-$\omega$ shear-stress transport two-equation model [32] calculated the turbulent viscosity. The pressure-velocity coupling executed the SIMPLE algorithm. The diffusion and convection terms adopted the central difference and second-order upwind discrete scheme, respectively [33].

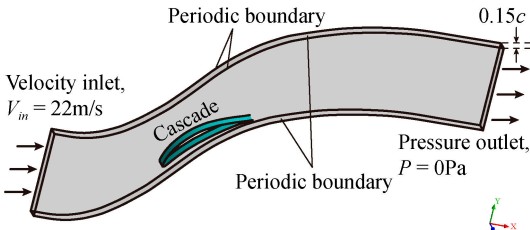

**Figure 3.** Computational domain and boundary conditions.

The LES is a turbulence method intermediate between direct numerical simulation and RANS turbulence modeling. Compared with the RANS calculation, the LES can get a more detailed description of the vortex structures in the cascade. The author's previous article explains the turbulence model and numerical method in detail [28]. The LES of relevant cases was carried out to obtain the influence of splitter blades on the vortex structure and pressure fluctuation in the cascade. The converged RANS results are taken as the initial flow field of LES. The dimensionless time step $\Delta t\, U_{in}/c = 0.01$, which is $7.27 \times 10^{-5}$ in this work, was adopted [2]. The total computational time steps were 10,000, and the corresponding calculation time ($T$) was 100.

*3.2. Grid Sensitivity Study and Validations*

For the base case, H-O-H multiblock structured grid topology was adopted. The inlet, outlet, and cascade channel adopted H-grid. The O-grid was applied around the blade. For the case with splitter blades, an O-grid topology was nested inside the H-grid topology of the cascade channel. The two grid topologies are shown in Figure 4. Due to the structural similarity, the same grid topology was adopted for different cases with splitter blades.

Figure 5 shows that five grid models of the base case and B3 were constructed, respectively, for a grid sensitivity study. The calculation results were conducted on RANS simulations. The predicted value and change rate of the static pressure coefficient ($C_p$) gradually decreased as the grid resolution increased. With the grid number of the base case reaching 8.13 million and B3 reaching 9.60 million, the predicted value and change rate of $C_p$ tended to be almost constant. Considering the computational economy and time cost, these two grid models were selected for subsequent calculations.

Table 3 shows the grid node distribution in the computational grid model's chordwise, normal, and spanwise directions. For the case with splitter blades, the grid node was mainly increased in the chordwise and normal directions. The $y+$ distribution of the blades in B3 at the DAA is shown in Figure 6. Most of the $y+$ on the main blade and the splitter blades were within the range of less than 1, which meets the numerical models' requirements. The results of the RANS were compared with the experiments in reference [29] to verify the accuracy of the simulation. The corresponding experimental setup can be found in

reference [29]. Figure 7 shows the $C_{ps}$ on the suction surface of the base case at the DAA. The predicted results are in good agreement with the experimental results, and it is considered that the grid model and numerical method used in this study are appropriate.

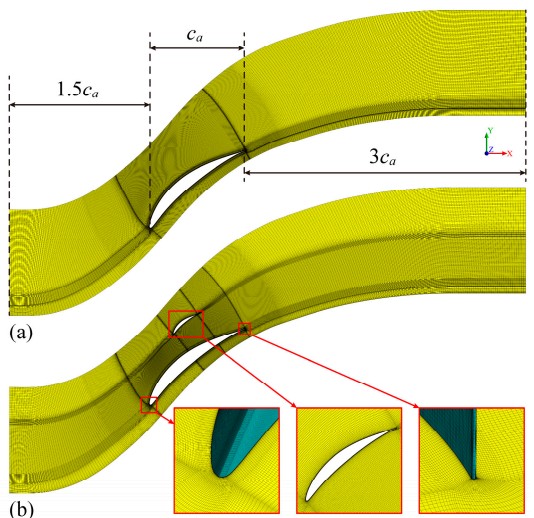

**Figure 4.** Distributions of grids: (**a**) base case; (**b**) B3.

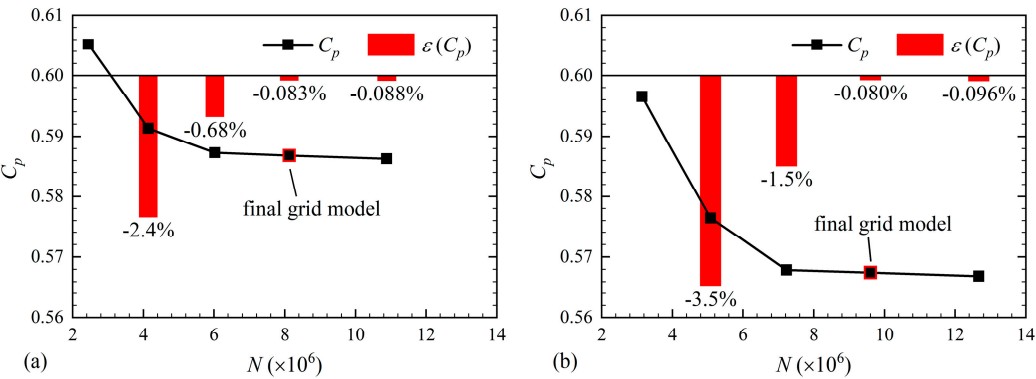

**Figure 5.** Grid sensitivity analysis: (**a**) base case; (**b**) B3.

**Table 3.** Grid node distributions of the computational grid model.

| Case | Grid Quantity ($\times 10^6$) | Number of Nodes (Chordwise) | Number of Nodes (Normal) | Number of Nodes (Spanwise) |
|---|---|---|---|---|
| Base case | 8.13 | 634 | 213 | 55 |
| B3 | 9.60 | 678 | 236 | 55 |

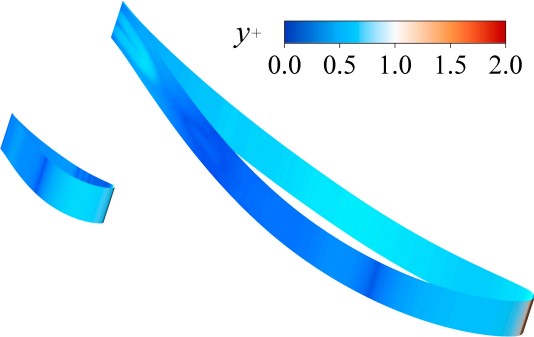

**Figure 6.** Contour plot of *y*+ distribution on the blade surface (B3, DAA).

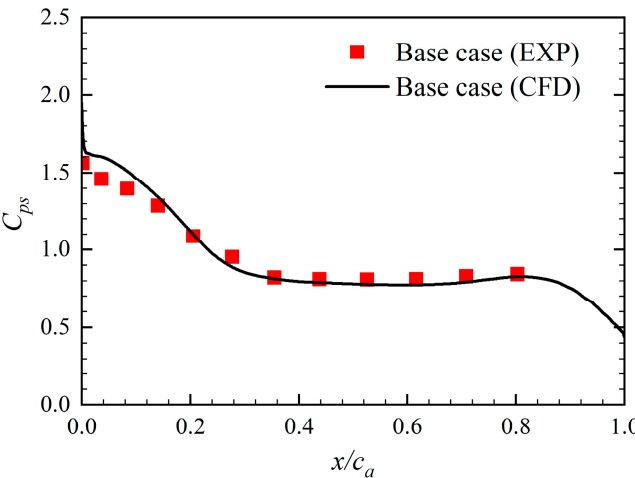

**Figure 7.** $C_{ps}$ of the experimental and predicted.

## 4. Results and Discussions

### 4.1. Effect of Splitter Blade Arrangement

The aerodynamic performance of the diffuser cascade is affected by the splitter blade arrangement [23,26]. The RANS ignores the unsteady state effects of the cascade flow field after stall. Therefore, the simulation started at the DAA when calculating the steady flow field. Then gradually the attack angle was increased until the $C_p$ reached the maximum value. This operating condition was regarded as the stall state of the cascade, and the calculation of larger attack angles was stopped.

Figure 8 shows the effect of the splitter blade arrangement on the $C_p$, total pressure loss ($C_{pt}$), and flow turning angle ($\Delta\beta$) of the diffuser cascade. Among them, the distance between the splitter blade and the suction surface of the main blade had a particularly significant impact on the aerodynamic performance of the diffuser cascade. When the splitter blade was close to the suction surface of the main blade ($J/s = 0.4$), the $C_p$ of the cascade with splitter blades decreased, and the $C_{pt}$ increased significantly. Meanwhile, the SAA of cascade with splitter blades was reduced, especially in the case near the leading edge of the main blade. These facts indicate that the splitter blades close to the suction surface will harm the internal flow control of the cascade. As the splitter blade was far away from the suction surface of the main blade, its negative impact on the internal flow of the cascade was weakened. When the splitter blade was arranged at the leading edge of the main blade ($I/c = 0$), the aerodynamic performance of the cascade was improved. According to the comprehensive analysis, the splitter blade should be arranged in the middle of the main blades near the leading edge ($I/c = 0$, $J/s = 0.5$), corresponding to the B1 in Figure 2, which can reach the best control on the internal flow of the diffuser cascade. The B1 had a higher $C_p$ and $\Delta\beta$ than the base case in the overall attack angle range. In addition, its SAA was 28°, which increases the stability of operation. Contrarily, the $C_{pt}$ of B1 reached its peak before the SAA and then decreased gradually. In the analysis of the following chapters, the flow separation mechanism controlled by splitter blades is explored by comparing the flow characteristics of the base case and B1.

### 4.2. Effect of Splitter Blades on the Flow Field

The comparison of the $C_p$ between the base case and B1 at the DAA and SAA is shown in Figure 9. Under the two attack angles, the scope of the low-pressure area near the leading edge of the B1 main blade was significantly smaller than that of the base case. A small low-pressure area appeared on the suction surface of the B1 splitter blades. As a result, the reverse pressure gradient on the suction surface of the B1 main blade was reduced, which is conducive to restraining the flow separation on the suction surface of the main blade. It can be observed that the splitter blades accelerated the energy conversion speed of the

entire system. The base case's $C_p$ became stable at about 1/3 chord length after the main blade. However, for B1, the $C_p$ was constant at the trailing edge of the main blade.

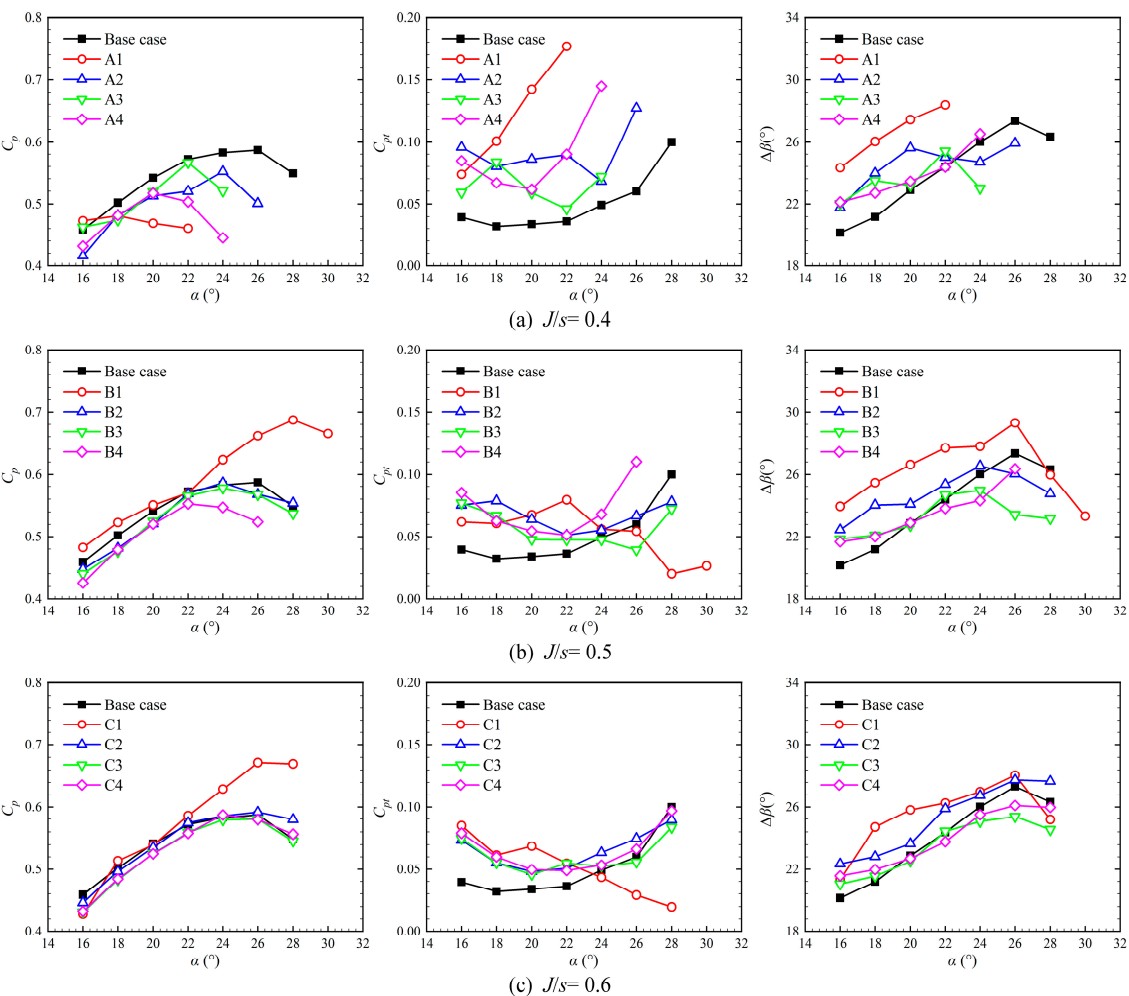

**Figure 8.** Effect of splitter blade arrangement on the $C_p$, $C_{pt}$, and $\Delta\beta$.

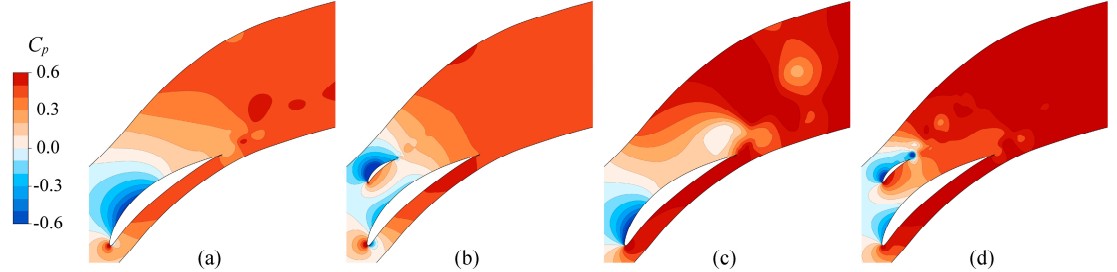

**Figure 9.** The $C_p$ distribution of (**a**) base case, DAA; (**b**) B1, DAA; (**c**) base case, SAA; (**d**) B1, SAA.

The $C_p$ distribution on the main blade surface of the base case and B1 is shown in Figure 10. In the figure, the blue dotted line represents the contour line of the main and splitter blades. The $C_p$ on the main blade changed obviously under the influence of the splitter blade. At the DAA, the $C_p$ of the main blade suction surface from the leading edge to 40% of the axial chord length was increased by arranging the splitter blades. The $C_p$ of the trailing edge also increased to a certain extent. At the SAA, the influence of splitter blades on the main blade suction surface was more significant. The $C_p$ of the main blade suction surface increased throughout the axial chord length, especially at the trailing edge.

The splitter blades reduced the load distribution of the main blade, indicating that the strength and scope of the separation vortex on the main blade was also weakened.

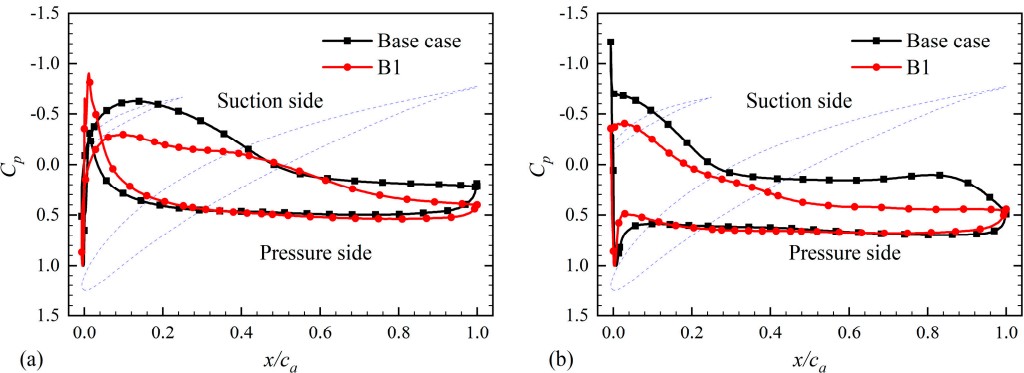

**Figure 10.** The $C_p$ comparison on the main blade surface at the (**a**) DAA; (**b**) SAA.

As an index to measure the development and decline of turbulence, turbulence kinetic energy can effectively evaluate the development and scale of the separated flow in the cascade system. Figure 11 compares turbulent kinetic energy between the base case and B1. At the DAA, along the chord direction, the flow in the base case gradually worsened, and the turbulent kinetic energy of the flow field gradually increased. Near the trailing edge, obvious flow separation vortices formed a high turbulent flow energy area. As the attack angle increased to the SAA, the base case began to form flow separation near the leading edge and covered more than half of the main blade suction surface. There were large vortices in the downstream flow field. After the splitter blades were arranged, the area of high turbulent kinetic energy on the main blades was significantly reduced, indicating that the flow separation was weakened. The main flow separation in the cascade occurred at the splitter blade suction surface. However, the splitter blade was significantly smaller than the main blade, so the total turbulent kinetic energy of B1 was smaller than the base case.

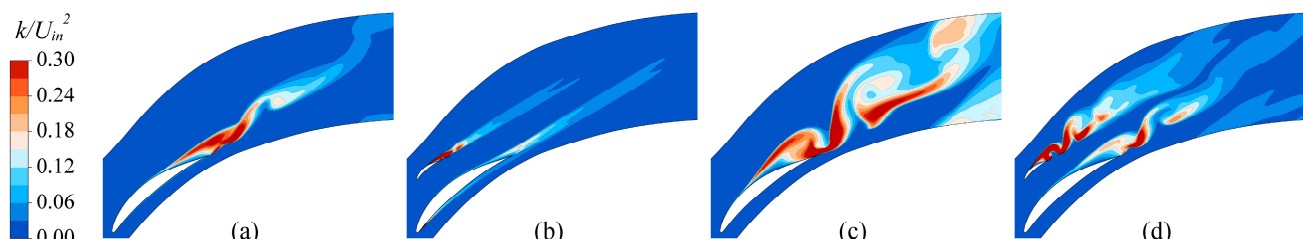

**Figure 11.** The turbulent kinetic energy of (**a**) base case, DAA; (**b**) B1, DAA; (**c**) base case, SAA; (**d**) B1, SAA.

The analysis mentioned above indicates that the splitter blade delayed flow separation on the main blade. Figure 12 compares boundary layer separation between the base case and B1 at the DAA and SAA. The delay in boundary layer separation led to a decrease in vortices near the blade surface [34,35]. The function of the splitter blades tends to stabilize the flow near the boundary layer of the main blade while suppressing the formation of surface vortices. However, the surface of the splitter blades forms small-scale vortices. This is beneficial for the stability of the flow in the cascade channel. Figure 13 shows the flow separation point's position of the base case and B1 at different attack angles. The flow separation point's position was linear with the attack angle. As the attack angle increased, the flow separation point moved towards the leading edge. All the separation points of B1 were about 0.2 axial chord length behind the base case.

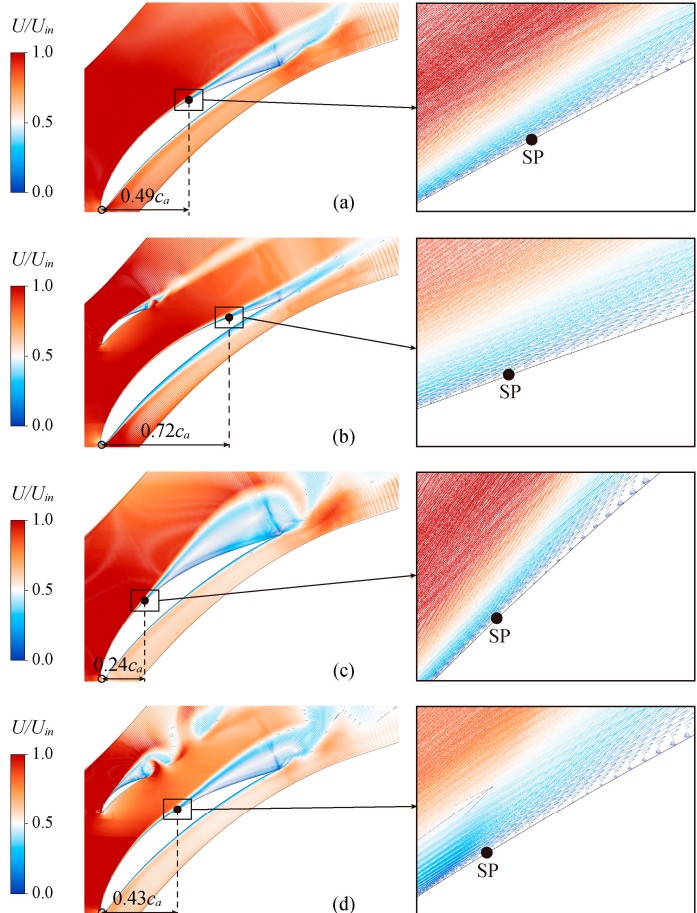

**Figure 12.** Effect of splitter blades on boundary layer separation: (**a**) base case, DAA; (**b**) B1, DAA; (**c**) base case, SAA; (**d**) B1, SAA.

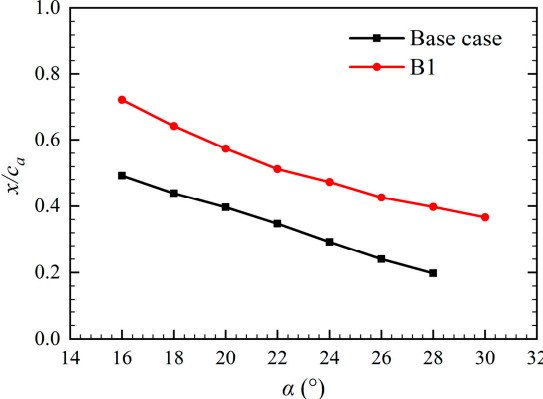

**Figure 13.** Position of flow separation point at different attack angles.

### 4.3. Effect of Splitter Blades on the Instantaneous Flow Structures

The LES computation was performed for the base case and B1 at the DAA and SAA to obtain the transient flow structures in the cascade. Figure 14 presents the comparison of transient $C_p$, $C_{pt}$, and $\Delta\beta$. The aerodynamic parameters fluctuation amplitude of B1 was lower than that of the base case, indicating that the splitter blade reduced the instability of the cascade flow field. The fluctuation amplitude of B1 aerodynamic parameters at the SAA was smaller than the base case at the DAA.

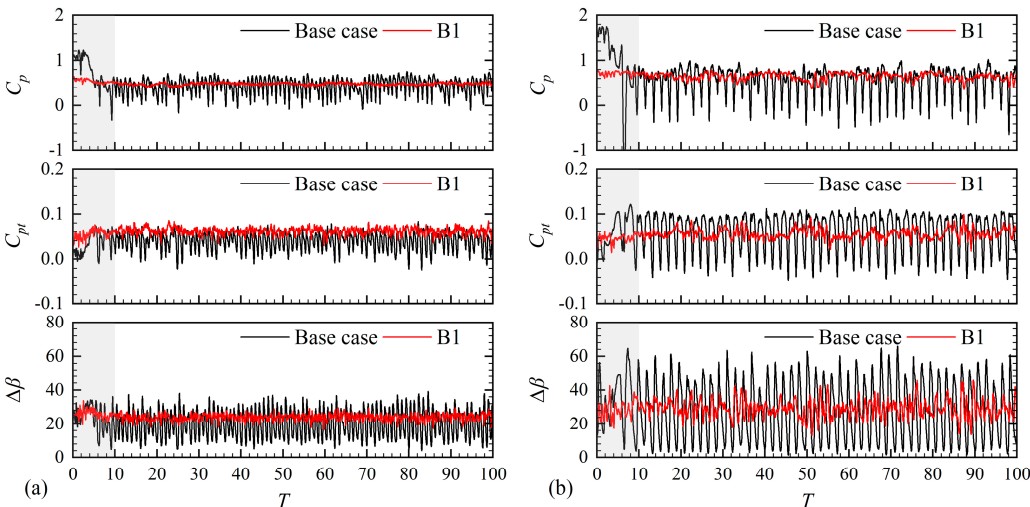

**Figure 14.** Comparison of transient $C_p$, $C_{pt}$, and $\Delta\beta$ at the (**a**) DAA; (**b**) SAA.

Figure 15 presents the curve of $C_p$ with time when $T$ was between 99 and 100. The $C_p$ of the base case had periodic characteristics at the DAA and SAA. The $C_p$ valley ($T_1$) and peak ($T_2$) moments of the base case at the DAA were selected for further analysis.

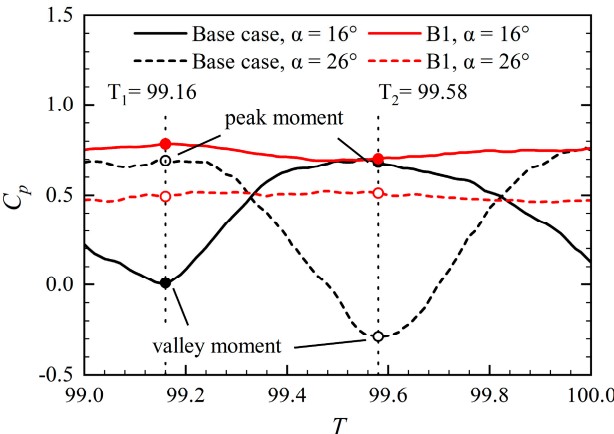

**Figure 15.** Static pressure coefficient when $T$ was 99~100.

To identify the three-dimensional vortex structure characteristics in the cascade at $T_1$ and $T_2$, the $Q$ criterion [36] was adopted to visualize the vortex structures:

$$Q = -\frac{1}{2}(S_{ij}S_{ij} - \Omega_{ij}\,\Omega_{ij}) \tag{3}$$

where

$$S_{ij} = \frac{1}{2}\left(\frac{\partial u_i}{\partial x_j} + \frac{\partial u_j}{\partial x_i}\right),\ \Omega_{ij} = \frac{1}{2}\left(\frac{\partial u_i}{\partial x_j} - \frac{\partial u_j}{\partial x_i}\right) \tag{4}$$

The internal transient vortex structures of the cascade at the DAA are shown in Figure 16. The vortex structures are identified by the $Q$ iso-surfaces, where $Q = 5 \times 10^5$, and colored by dimensionless velocity. The control mechanism of splitter blades can be observed by comparing the vortex structures of the base case and B1. For the base case, the two-dimensional shear layers generated at the leading edge rapidly burst into three-dimensional vortices, forming a large-scale separation region on the suction surface. Large-scale vortices were observed in the downstream flow field of the cascade. The periodic shedding of vortices from the suction surface of the main blade can cause periodic pressure fluctuations in the cascade channel, reducing the stability of the flow inside the cascade. The splitter

blade changed the generation and development of the vortex structures on the main blade. The arrangement of the splitter blades significantly delayed the separation region of the main blade and reduced the size and scope of the separation vortex. This is the reason why the $C_p$ of the cascade with splitter blades increased. The separation vortices generated on the splitter blade and the main blade rapidly decomposed into small-scale vortices downstream of the cascade and gradually disappeared. Therefore, the pressure fluctuation inside the cascade channel was weakened. This thoroughly explains the mechanism of splitter blades to increase the flow stability of the cascade.

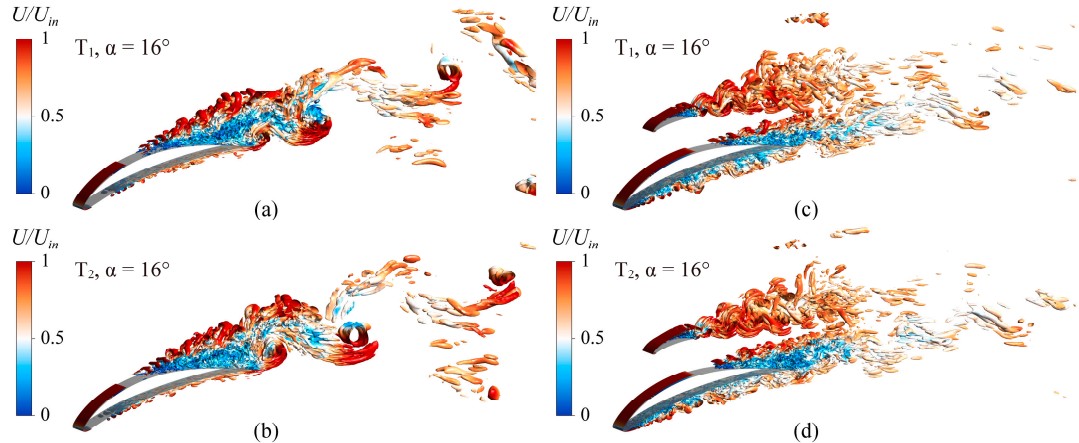

**Figure 16.** The vortex structures of the cascade flow field at the DAA, identified by the iso-surfaces of $Q = 5 \times 10^5$ colored with velocity: (**a**) base case, $T_1$; (**b**) base case, $T_2$; (**c**) B1, $T_1$; (**d**) B1, $T_2$.

Figure 17 shows the internal transient vortex structures of the cascade at the SAA. The separation region on the base case blade extended to the leading edge, and the scale of the separation vortices increased significantly, resulting in a rapid increase in $C_{pt}$. The base case reached a near-stall flow state. For B1, although the scale of the separation vortices on the splitter blade had increased, their intensity had decreased. The separation point of the main blade extended towards the trailing edge, reducing the size and intensity of the separation vortex on the suction surface. The separation vortices of the splitter blade and the main blade interacted and weakened downstream of the cascade. Therefore, the $C_{pt}$ of B1 at the SAA was reduced. The enhanced internal flow stability of B1 delayed the occurrence of a near-stall state.

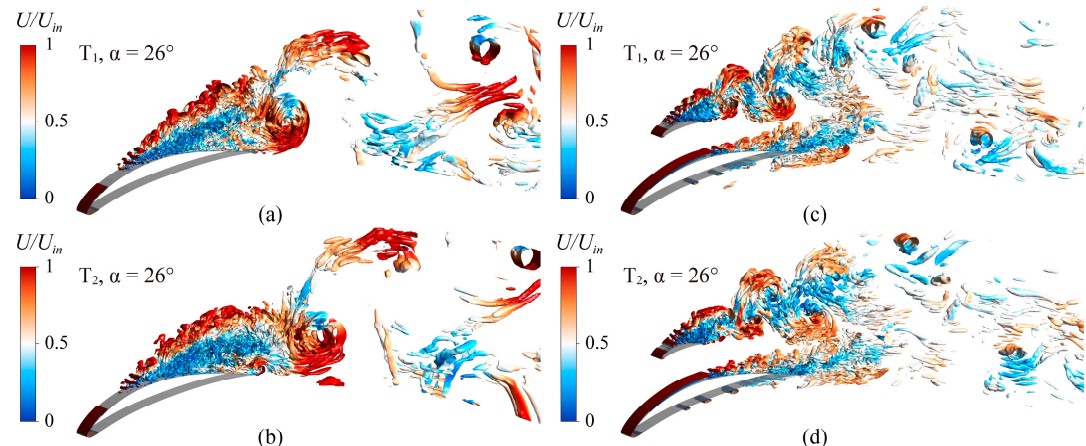

**Figure 17.** The vortex structures of the cascade flow field at the SAA, identified by the iso-surfaces of $Q = 5 \times 10^5$ colored with velocity: (**a**) base case, $T_1$; (**b**) base case, $T_2$; (**c**) B1, $T_1$; (**d**) B1, $T_2$.

*4.4. Effect of Splitter Blades on the Pressure Fluctuation*

The vortex structures in Figures 16 and 17 indicate that the evolution process of the separated vortex on the cascade with and without splitter blades was entirely different. The separation vortices on the splitter blade and the main blade are also different from each other.

For the base case, a monitor was set at the trailing edge of the blade. For B1, two monitors were set in the flow field, respectively located at the trailing edge of the main and splitter blades. Their locations are shown in Figure 18. The evolution and fluctuation law of the separation vortex was explored by monitoring the $C_p$ fluctuation at the trailing edge of the blade.

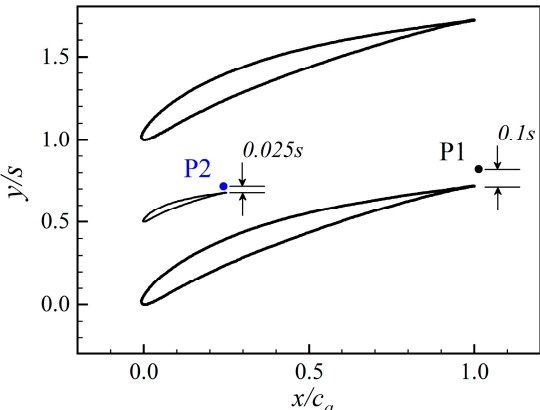

**Figure 18.** Location of the monitors at the trailing edge: P1—main blade monitor; P2—splitter blade monitor.

Figure 19 shows the pressure fluctuation and its corresponding fast Fourier transform (FFT) spectrum of monitors at the DAA. The results indicate that the arrangement of the splitter blades weakened the pressure fluctuation on the main blade and improved the flow stability. However, the separation vortex on the splitter blade fluctuated. The spectrum demonstrates that the separation vortices of base case blades and B1 splitter blades had prominent peaks, while the separation vortices of B1 main blades did not.

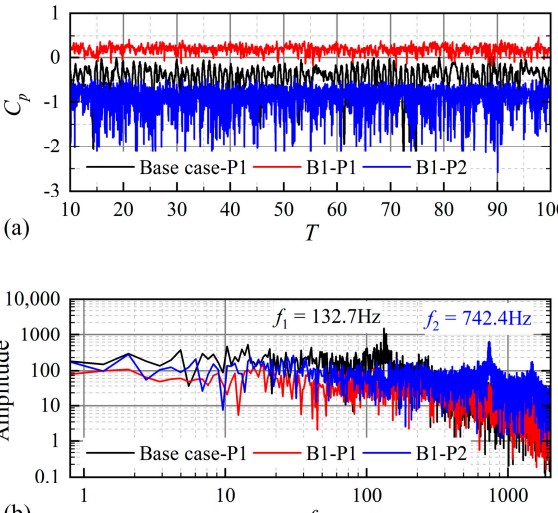

**Figure 19.** Pressure fluctuations of monitors at the DAA: (**a**) the $C_p$; (**b**) the corresponding FFT spectrum.

The pressure fluctuation and corresponding FFT spectrum at the SAA are shown in Figure 20. The pressure fluctuation amplitude was strengthened for the base case, and the peak frequency decreased to 72.2 Hz. For B1, the $C_p$ on the main blade increased. Moreover,

the separation vortex's fluctuation on the splitter blade was weaker, manifested by the weakening pressure fluctuation and the disappearance of peak frequency at the monitor. This is consistent with the vortex structures in Figure 17.

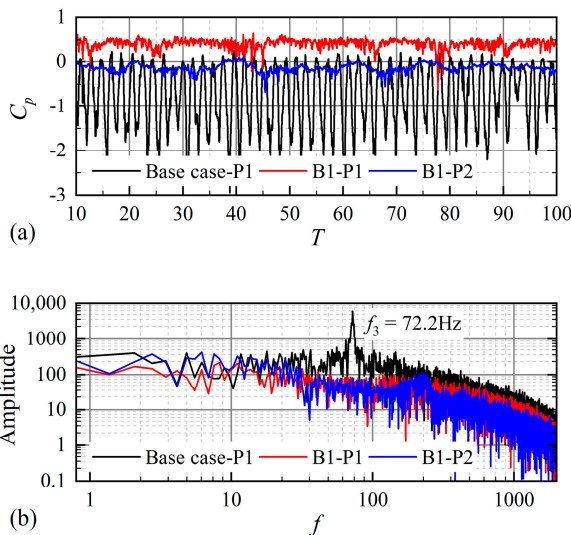

(a)

(b)

**Figure 20.** Pressure fluctuations of monitors at the SAA: (**a**) the $C_p$; (**b**) the corresponding FFT spectrum.

## 5. Conclusions

The passive flow control technology of flow separation in low-speed diffuser cascade was investigated in this paper. The effects of splitter blades on the aerodynamic performance and internal unsteady flow characteristics of low-speed diffuser cascade were studied by the numerical method. The influences of different splitter blade arrangement positions on the aerodynamic performance of the cascade were investigated in detail by RANS calculations, and the optimal arrangement position was determined. Then, the LES simulations of the base case and the optimized splitter blade case were further conducted to obtain the transient vortex structures and unsteady flow characteristics of the cascade. Moreover, the flow control mechanism of the splitter blade in the low-speed compressor cascade was explained. The flow control mechanism of the splitter blades in the low-speed diffuser cascade was expounded. The main results derived from this study are as follows:

1.  The aerodynamic performance of the low-speed diffuser cascade was susceptible to the position of the splitter blades. The optimal position of the splitter blade was located in the middle of the main blades near the leading edge (B1, $I/c = 0$, $J/s = 0.5$). The aerodynamic performance of the cascade system was effectively improved.

2.  The arrangement of the splitter blades changed the pressure distribution on the main blades. Decreasing the reverse pressure gradient on the main blade suction surface weakened the separation flow. The main separation flow occurred on the splitter blade suction surface. All the separation points of B1 were about 0.2 axial chord length behind the base case.

3.  The LES results show that the splitter blades improved the stability of the cascade system, especially at the stall angle of attack. With the arrangement of splitter blades, the scale and intensity of the separation vortices generated on the suction surface of the main blade decreased. In addition, the separation vortices of the main blade and the splitter blade interacted and rapidly decomposed into small-scale vortices downstream of the cascade, reducing the flow loss.

**Author Contributions:** Conceptualization, Z.L., J.W. and W.Y.; methodology, Z.L., B.J. and H.Z.; software, Z.L., B.J. and J.L.; validation, B.J., W.Y. and J.L.; formal analysis, Z.L., J.W. and W.Y.; investigation, Z.L., H.Z. and J.L.; resources, Z.L., W.Y. and J.L.; data curation, H.Z. and B.J.; writing—original draft preparation, Z.L. and J.W.; writing—review and editing, Z.L., J.W. and B.J.; visualization, Z.L. and H.Z; supervision, J.W. and W.Y.; project administration, Z.L., H.Z. and W.Y.; funding acquisition, Z.L. and J.W. All authors have read and agreed to the published version of the manuscript.

**Funding:** This study was supported by project funded by the China Postdoctoral Science Foundation (No. 2022M721238).

**Data Availability Statement:** The data presented in this study are available on request from the corresponding author. The data are not publicly available due to the data occupying too much memory.

**Acknowledgments:** The authors acknowledge the support of the China Postdoctoral Science Foundation (Grant No. 2022M721238) and all the other scholars for their advice in the process of improving this article. The computation was completed in the HPC Platform of Huazhong University of Science and Technology.

**Conflicts of Interest:** Weigang Yang and Jieda Ling were was employed by the company Ningbo Fotile Kitchen Ware Corp. Ltd., the remaining authors declare that the research was conducted in the absence of any commercial or financial relationships that could be construed as a potential conflict of interest.

## Nomenclature

| | |
|---|---|
| $C_p$ | Static pressure coefficient, $C_p = (P_{s,out} - P_{s,in})/(P_{t,in} - P_{s,in})$ |
| $C_{ps}$ | Blade surface pressure coefficient, $C_{ps} = (P_{t,in} - P_s)/(P_{t,in} - P_{s,in})$ |
| $C_{pt}$ | Total pressure loss coefficient, $C_{pt} = (P_{t,in} - P_{t,out})/(P_{t,in} - P_{s,in})$ |
| $\varepsilon(C_p)$ | Static pressure coefficient variable rate, $\varepsilon(C_p) = (C_{p,i} - C_{p,i-1})/(C_{p,i-1}) \times 100\%$ |
| $\Delta\beta$ | Flow turning angle, ° |
| $U_{in}$ | Flow velocity at the cascade inlet, m/s |
| $P_s$ | Static pressure, Pa |
| $P_t$ | Total pressure, Pa |
| $c$ | Blade chord length, mm |
| $c_a$ | Blade axial chord length, mm |
| $s$ | Pitch spacing, mm |
| $\varphi$ | Camber angle, ° |
| $\gamma$ | Stagger angle, ° |
| $h$ | Blade span, mm |
| $i$ | Incidence angle, ° |
| $\alpha$ | Attack angle, ° |
| $\beta_1$ | Design inflow angle, ° |
| $\beta_2$ | Design outflow angle, ° |
| $T$ | The dimensionless time |
| $\Delta t$ | Physical time step, s |
| $T_1$ | Valley moment, s |
| $T_2$ | Peak moment, s |
| P1 | Main blade monitor |
| P2 | Splitter blade monitor |
| DAA | Design attack angle |
| SAA | Stall attack angle |
| RANS | Reynolds averaged Navier-Stokes |
| LES | Large-eddy simulation |
| FFT | Fast Fourier transform |

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
