# Peer review of "Large-Eddy Simulation of Flow Separation Control in Low-Speed Diffuser Cascade with Splitter Blades"

_processes, doi:10.3390/pr11113249_

Round 1
Reviewer 1 Report
Comments and Suggestions for Authors
Dear authors!
The aim of the paper is clear, and the results seems to be of practical interest.
But there are few unsignificant lacks in the paper.
1. I couldn't find the meaning of epsilon(C_p) in the text (see Fig. 5).
2. References for Fig. 7 and Fig. 8 are incorrect in the text (page 7): there should be Fig. 6 instead of Fig. 7 and Fig. 7 instead of Fig. 8.
3. In captions of Fig. 9 and Fig. 11: (a) and (b) comes twice instead of (a), (b),(c), and (d)
Author Response
Response to Reviewer 1 Comments
Point 1: I couldn't find the meaning of epsilon(C_p) in the text (see Fig. 5).
Response 1: Thank you for your valuable comments on the manuscript. The author added the equation for static pressure coefficient variable rate ε(Cp) in the Nomenclature. The ε(Cp) is defined to evaluate the convergence of the static pressure coefficient Cp with the increase in the number of grid cells N:
ε(Cp) = (Cp,i - Cp,i-1 )/( Cp,i-1)×100%
where Cp,i is the static pressure coefficient calculated by the i-th grid model.
Point 2: References for Fig. 7 and Fig. 8 are incorrect in the text (page 7): there should be Fig. 6 instead of Fig. 7 and Fig. 7 instead of Fig. 8.
Response 2: The author has corrected the references to Figures 6 and 7.
Point 3: In captions of Fig. 9 and Fig. 11: (a) and (b) comes twice instead of (a), (b),(c), and (d).
Response 3: The author has corrected the errors in the captions of Figures 9 and 11.

Reviewer 2 Report
Comments and Suggestions for Authors
The authors present LES study splitter blades on the flow field and aerodynamic performance. The paper is well written.
I would suggest authors make captions more descriptive and mention if the results are RANS-based or LES-based
Another suggestion is to make sure to explain what more features LES captures that RANS fail to capture.
It is not apparent immediately if the grid convergence studies are conducted on RANS simulations or LES simulations. The authors should clarify that.
I just want to draw attention to works on LES of slender bodies like spheroids already done in the literature and cite them appropriately in the paper.
1. The high-Reynolds-number stratified wake of a slender body and its
comparison with a bluff-body wake JL Ortiz-Tarin, S Nidhan, S Sarkar
Journal of Fluid Mechanics 957, A7
2.High-Reynolds-number wake of a slender body JL Ortiz-Tarin, S Nidhan,
S Sarkar Journal of Fluid Mechanics 918, A30
3. Large-eddy simulation of tripping effects on the flow over a 6 : 1 prolate spheroid at angle of attack M Plasseraud, P Kumar, K Mahesh Journal of Fluid Mechanics 960, A3
4. Large-eddy simulation of flow over an axisymmetric body of revolution P Kumar, K Mahesh Journal of Fluid Mechanics 853, 537-563
Comments on the Quality of English Language
good english, please do some editing before submission
Author Response
Response to Reviewer 2 Comments
Point 1: I would suggest authors make captions more descriptive and mention if the results are RANS-based or LES-based.
Response 1: Thank you for your valuable comments on the manuscript. The author has adjusted the captions, highlighting that the results are LES-based.
Point 2: Another suggestion is to make sure to explain what more features LES captures that RANS fail to capture.
Response 2: The author further explained the vortex structure features captured by LES in Figures 16 and 17.
Point 3: It is not apparent immediately if the grid convergence studies are conducted on RANS simulations or LES simulations. The authors should clarify that.
Response 3: The grid convergence studies are conducted on RANS simulations. The author added relevant explanations in the middle of section 3.2 (Page 7).
Point 4: I just want to draw attention to works on LES of slender bodies like spheroids already done in the literature and cite them appropriately in the paper.
- The high-Reynolds-number stratified wake of a slender body and its comparison with a bluff-body wake. JL Ortiz-Tarin, S Nidhan, S Sarkar Journal of Fluid Mechanics 957, A7
2.High-Reynolds-number wake of a slender body. JL Ortiz-Tarin, S Nidhan, S Sarkar Journal of Fluid Mechanics 918, A30
- Large-eddy simulation of tripping effects on the flow over a 6 : 1 prolate spheroid at angle of attack. M Plasseraud, P Kumar, K Mahesh Journal of Fluid Mechanics 960, A3
- Large-eddy simulation of flow over an axisymmetric body of revolution. P Kumar, K Mahesh Journal of Fluid Mechanics 853, 537-563
Response 4: The papers recommended by the reviewers are very precious. The author cited these papers in the manuscript.

Reviewer 3 Report
Comments and Suggestions for Authors
This paper studies the passive flow control technology of using splitter blades in low-speed diffuser cascade. Overall, the results are interesting and could be considered for publication after major revision on the numerical analysis part:
1) Introduction. I suggest to include splitter blade diagram here, and to include important items for ease of readership near the line 92.
2) Section 3.1. Please include a table of boundary conditions with mathematical descriptions
3) Figure 7. Experimental value is presented but no experimental setup is included. This is a major revision that is needed.
4) Figure 12. More details are needed on the separation point. Please refer to these suggested references on the separation angle that discusses impact of separation point on instability
i) doi.org/10.1016/j.memsci.2021.119599
ii) doi.org/10.1016/j.oceaneng.2023.114932
5) Figure 14 is unclear. Please improve resolution
6) Figure 16 and 17. Please indicates with some wordings on flow instability
Author Response
Response to Reviewer 3 Comments
Point 1: Introduction. I suggest to include splitter blade diagram here, and to include important items for ease of readership near the line 92.
Response 1: Thank you for your valuable comments on the manuscript. The splitter blade diagram is already included in Section 2.2(Page 5), and important parameters for the splitter blades are introduced. If a splitter blade diagram is added to the Introduction, it would be a repetition of the content in Section 2.2.
Point 2: Section 3.1. Please include a table of boundary conditions with mathematical descriptions.
Response 2: The boundary conditions are described in section 3.1. The boundary conditions of the computational domain are relatively simple, divided into velocity inlet, pressure outlet, and periodic boundary conditions. The airflow velocity at the inlet is 22m/s, which can be calculated based on Ma in Figure 1. The pressure at the outlet is set to 0Pa. The author has added the corresponding mathematical description to section 3.1 and new figure 3.
Point 3: Figure 7. Experimental value is presented but no experimental setup is included. This is a major revision that is needed.
Response 3: The experimental value in this manuscript is from reference [29]. The author did not conduct any corresponding experiments, and the related experimental settings can be found in reference [29]. The author provided additional explanations.
Point 4: Figure 12. More details are needed on the separation point. Please refer to these suggested references on the separation angle that discusses impact of separation point on instability
- i) doi.org/10.1016/j.memsci.2021.119599
- ii) doi.org/10.1016/j.oceaneng.2023.114932
Response 4: The author has updated Figure 12 and added more details on separation points. The discussion of separation points on flow instability has been added in section 4.2 (Page 12).
Point 5: Figure 14 is unclear. Please improve resolution.
Response 5: The author has replaced the higher resolution figure with 1200DPI
Point 6: Figure 16 and 17. Please indicates with some wordings on flow instability.
Response 6: The author has added a description of flow instability.

Round 2
Reviewer 3 Report
Comments and Suggestions for Authors
The authors have addressed all the comments. Accept as it is.